# Perforin inhibition protects from lethal endothelial damage during fulminant viral hepatitis

M. Welz[1], S. Eickhoff[1], Z. Abdullah[1], J. Trebicka [2,3], K.H. Gartlan [4], J.A. Spicer[5], A.J. Demetris[6], H. Akhlaghi[7], M. Anton[8], K. Manske[8], D. Zehn[9], B. Nieswandt[10], C. Kurts[1], J.A. Trapani[7], P. Knolle[8], D. Wohlleber [8] & W. Kastenmüller [1,11]

CD8 T cells protect the liver against viral infection, but can also cause severe liver damage that may even lead to organ failure. Given the lack of mechanistic insights and specific treatment options in patients with acute fulminant hepatitis, we develop a mouse model reflecting a severe acute virus-induced CD8 T cell-mediated hepatitis. Here we show that antigen-specific CD8 T cells induce liver damage in a perforin-dependent manner, yet liver failure is not caused by effector responses targeting virus-infected hepatocytes alone. Additionally, CD8 T cell mediated elimination of cross-presenting liver sinusoidal endothelial cells causes endothelial damage that leads to a dramatically impaired sinusoidal perfusion and indirectly to hepatocyte death. With the identification of perforin-mediated killing as a critical pathophysiologic mechanism of liver failure and the protective function of a new class of perforin inhibitor, our study opens new potential therapeutic angles for fulminant viral hepatitis.

[1] Institute of Experimental Immunology, University Hospital, University of Bonn, 53127 Bonn, Germany. [2] Department of Internal Medicine I, University of Bonn, Bonn 53127, Germany. [3] European Foundation for the Study of Chronic Liver Failure, Barcelona 08021, Spain. [4] QIMR Berghofer Medical Research Institute, 300 Herston Road QLD, Herston 4006, Australia. [5] Auckland Cancer Society Research Centre, University of Auckland, Auckland 92019, New Zealand. [6] Department of Pathology, University of Pittsburgh, Pittsburgh 15213 PA, USA. [7] Peter MacCallum Cancer Centre, Melbourne 3000 VIC, Australia. [8] Institute of Molecular Immunology and Experimental Oncology, Klinikum rechts der Isar, Technical University of Munich, München 81675, Germany. [9] Division of Animal Physiology and Immunology, School of Life Sciences Weihenstephan, Technical University of Munich, Weihenstephaner Berg 3, Freising 85354, Germany. [10] Institute of Experimental Biomedicine I, University of Wuerzburg, Wuerzburg 97078, Germany. [11] Institute of Systems Immunology, University of Wuerzburg, Wuerzburg 97078, Germany. These authors contributed equally: M. Welz, S. Eickhoff, P. Knolle, D. Wohlleber, W. Kastenmüller. Correspondence and requests for materials should be addressed to P.K. (email: percy.knolle@tum.de) or to W.K. (email: wolfgang.kastenmueller@uni-wuerzburg.de)

Major threats to human health on a global scale are infections with hepatotropic viruses, such as Hepatitis B virus (HBV), Hepatitis C virus, Hepatitis D virus, and Hepatitis E virus as well as parasitic infections like malaria[1,2]. The liver is known to regulate local as well as systemic immune responses through its unique immunological properties and tolerogenic antigen-presenting cell populations[3,4]. This tolerogenic function of the liver is considered to contribute to the development of persistent hepatitis virus infections by impairing effective immune protection[5,6]. Yet, most acute infections with Hepatitis virus A, B or E occurring during adulthood are cleared by CD8 T cell immunity[2], suggesting a well-balanced regulation between immunity and tolerance in the liver. Rarely, fulminant cases of viral hepatitis are observed after acute infection with hepatitis viruses[7] and strong (re)-activation of virus-specific immunity following rituximab treatment[8] or during the immune reconstitution inflammatory syndrome in HIV patients co-infected with Hepatitis B[9]. The development of immune-mediated liver failure during viral hepatitis demonstrates that despite its tolerogenic function the liver can become target of devastating antiviral immunity, for which currently no specific pharmacological therapy is available. Liver transplantation is therefore the only life-saving option available for deteriorating patients with acute fulminant hepatitis[10].

Several effector mechanisms that explain how CD8 T cells can cause severe hepatitis have been identified in preclinical models. Among them are cytokines like interferon (IFN)-γ and tumor necrosis factor (TNF) as well as the death effector molecules FASL and perforin-1[11–15]. Also a role for natural killer cells in severe viral hepatitis has been proposed[16–18]. Yet, it remains unknown which mechanisms are responsible for T cell-mediated liver failure in the context of, e.g., a fulminant Hepatitis B. In patients with fulminant hepatitis, very high numbers of immune cells are found in the liver and higher numbers of virus-specific effector CD8 T cells are detected compared to patients with acute hepatitis[19]. Virus-specific T cells in patients with fulminant hepatitis also showed increased IFN-γ expression[20] and lack of upregulation of co-inhibitory receptors such as PD1 on CD8 T cells correlated with disease progression[21]. This dual role of CD8 T cells in not only antiviral protection but also damage has been recognized many years ago[22], yet the molecular and cellular mechanisms that determine the outcome of CD8 T cell immunity for organ integrity remained unknown.

Here we set out to develop a new model for an acute fulminant CD8 T cell-dependent viral hepatitis in order to gain mechanistic insights regarding the critical effector function of CD8 T cells with the goal to develop new therapeutic angles to approach this severe condition. On a mechanistic level, we found that perforin-mediated killing was a critical function of antigen-specific CD8 T cells during fulminant hepatitis. Importantly, T cell-mediated hepatitis was dependent on direct killing of hepatocytes, but the development toward fulminance additionally required perforin-mediated elimination of liver sinusoidal endothelial cells (LSECs). This led to dramatic alterations of hepatic vascular perfusion and secondary hepatocyte death. Therapeutically, we were able to rescue animals during the onset of disease with a newly developed perforin-1 inhibitor, opening new potential avenues to treat patients with acute CD8 T cell-mediated liver failure.

## Results

**A model of CD8 T cell-mediated acute liver failure**. In order to characterize the pathophysiologically relevant mechanisms of CD8 T cell-induced liver failure during fulminant viral hepatitis, we first set out to develop a new mouse model. Specifically, we adoptively transferred physiological numbers ($1 \times 10^4$) of naive OT-I cells (ovalbumin (OVA)-specific, H-2K$^b$-restricted, T cell receptor (TCR) transgenic CD8 T cells) into wild-type (wt) recipient mice and vaccinated them with a combination of OVA protein, polyinosinic–polycytidylic acid (poly I:C) and αCD40-stimulating antibody as adjuvants (Fig. 1a) that leads to the formation of a robust T cell memory population by day 30 after vaccination[23]. Mice were then infected intravenously (i.v.) with recombinant adenoviruses coding for green fluorescent protein (GFP)/OVA/Luciferase (AdGOL) or GFP/Luciferase (AdGL) as a control, reflecting an acute viral infection of the liver. Within 3 days after infection, AdGOL-infected mice showed a strong increase in serum alanine transaminase (ALT) levels and loss of body weight (Fig. 1b, c). In contrast, in AdGL-infected control mice ALT levels and body weight remained unaltered. By day 4 after infection, 80% of vaccinated AdGOL-infected mice succumbed, while all animals survived in the control groups (Fig. 1d). In line with a key function of CD8 T cell immunity, we observed large numbers (up to $15 \times 10^6$) of OT-I T cells in the livers at the peak of the disease (Fig. 1e). In summary, this model reflects an acute CD8 T cell-mediated fulminant viral hepatitis.

Next we tested whether a fulminant hepatitis was also observed after generating memory T cells from the endogenous repertoire rather than using TCR transgenic T cells. To this end, we immunized mice (OVA protein/poly I:C/αCD40) and challenged them with AdGOL 30 days later (Fig. 1f). After vaccination, endogenous OVA-specific T cells also induced severe liver damage within 3 days after infection. Again we observed a steep increase in ALT levels and loss of body weight (Fig. 1g, h). Given the CD8 T cell dependence of this model, we assumed that the numbers of antigen-specific CD8 T cells in the blood before infection may correlate with the ALT serum levels on day 3 after infection. We detected a mild correlation (Fig. 1i), arguing that besides antigen-specific T cell numbers before infection also expansion and likely the presence of liver-resident T cells are critical factors regarding disease severity. Currently, it is unknown whether a fulminant hepatitis is caused by activated naive CD8 T cells or rather by cross-reactive memory CD8 T cells or a combination of both. In order to test the capacity of naive T cells to elicit a fulminant hepatitis, we transferred $7 \times 10^6$ CD44$^-$ (naive) OT-I T cells into recipient mice and infected them with AdGOL or AdGL as a control on the same day (Fig. 1j). Similar to the setting described above, we observed antigen-dependent lethality in ≈70% of the mice (Fig. 1k–m). If fewer OT-I T cells were transferred or less virus than $5 \times 10^8$ AdGOL were used for infection, severe liver damage but no fulminant viral hepatitis was observed (Supplementary Fig. 1) indicating a threshold for both the numbers of antigen-specific CD8 T cells and the number of infected hepatocytes beyond which fulminant hepatitis develops. In order to exclude that the observed pathology in our model was restricted to the OVA/OT-I system, we generated an adenovirus that expresses a viral epitope derived from murine lymphocytic choriomeningitis virus (LCMV; GP$_{33}$). Ad-GOL-GP$_{33}$-infected wt mice that received adoptively transferred GP$_{33}$-specific TCR transgenic CD8 T cells (P14) developed a fulminant hepatits leading to dramatic weight loss and death (Supplementary Fig. 2). With these data, we establish a preclinical mouse model that mimics a CD8 T cell-mediated fulminant hepatitis in humans.

Next, we analyzed the liver tissue of diseased mice on day 3 post AdGOL infection and adoptive T cell transfer to determine the localization of T cells within the infected livers. Immunofluorescent staining of liver sections revealed a massive and antigen-specific infiltration of CD8 T cells around periportal areas (Fig. 2a). In line with a CD8 T cell-mediated elimination of infected hepatocytes, we observed a loss of GFP signal (infected hepatocytes) around periportal areas, where CD8 T cells infiltrated into liver tissue (Fig. 2a). It is noteworthy that T cells

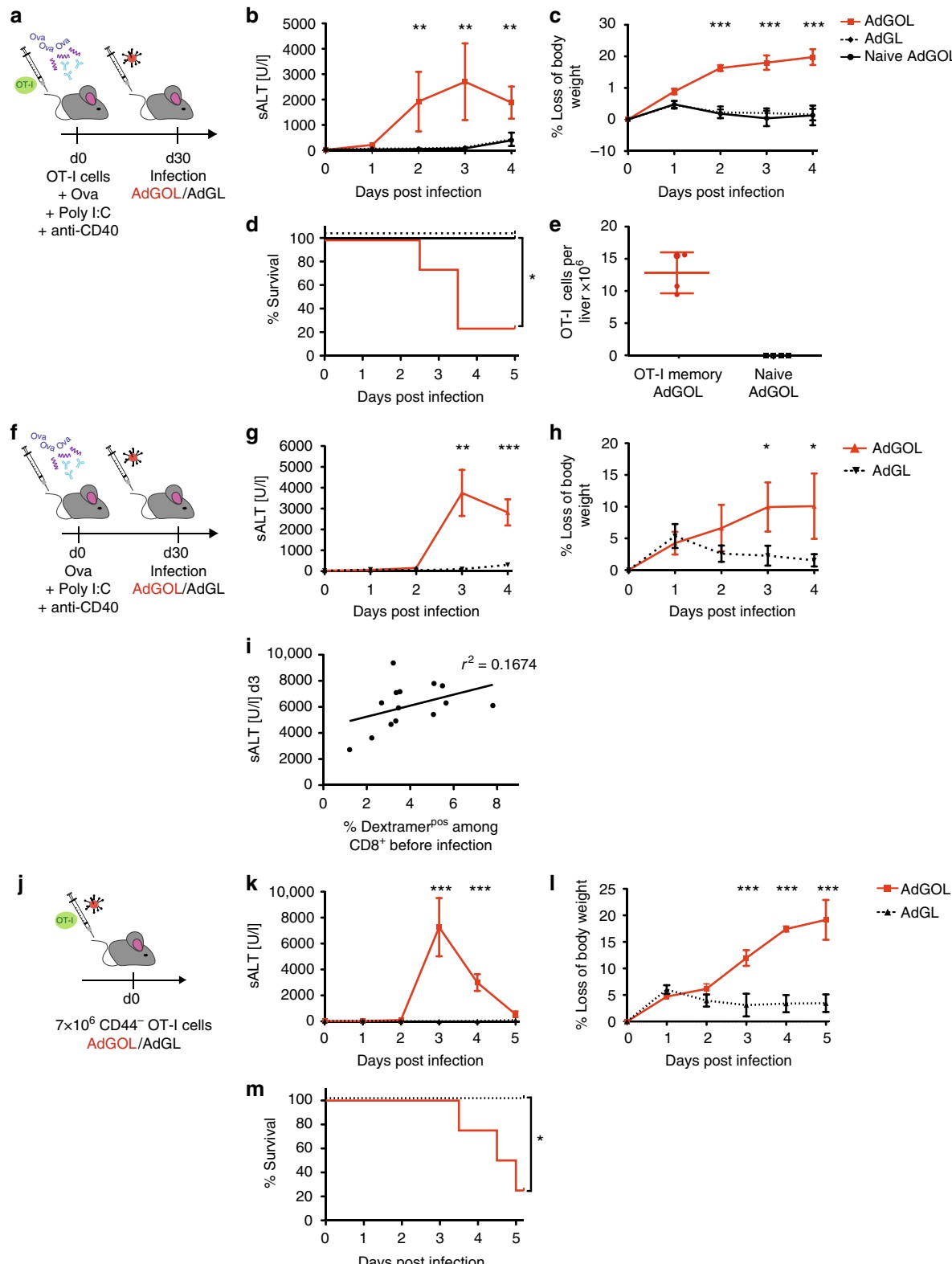

**Fig. 1** Model of CD8 T cell-mediated acute liver failure. **a**, **f**, **j** Schematics showing various approaches to induce a CD8 T cell-mediated hepatitis using OT-I memory T cells (**a**), endogenous OVA-specific memory T cells (**f**), or high numbers of acutely transferred naive OT-I T cells (**j**). **b**, **g**, **k** ALT levels in the serum. **c**, **h**, **l** The percentage of loss of body weight over time. **d**, **m** The survival curves of mice. **e** Quantification of (CD45.1) congenic OT-I cell in the livers of mice on day 3 post infection, gating strategy is shonw in Supplementary Fig. 1. **i** Correlation of antigen-specific CD8 T cells on day 30 post immunization with serum ALT activity on day 3 post infection (see approach shown in **f**). Data are representative of two (**f**–**i**) or three independent experiments (n = 4). Error bars indicate the mean ± SD. Comparison between groups was calculated using unpaired Student's t test (**g**, **h**, **k**, **l**), one-way ANOVA with a Tukey's multiple comparison post test (**b**, **c**) or log-rank test (**d**, **m**). *p ≤ 0.05; **p < 0.01; ***p < 0.001

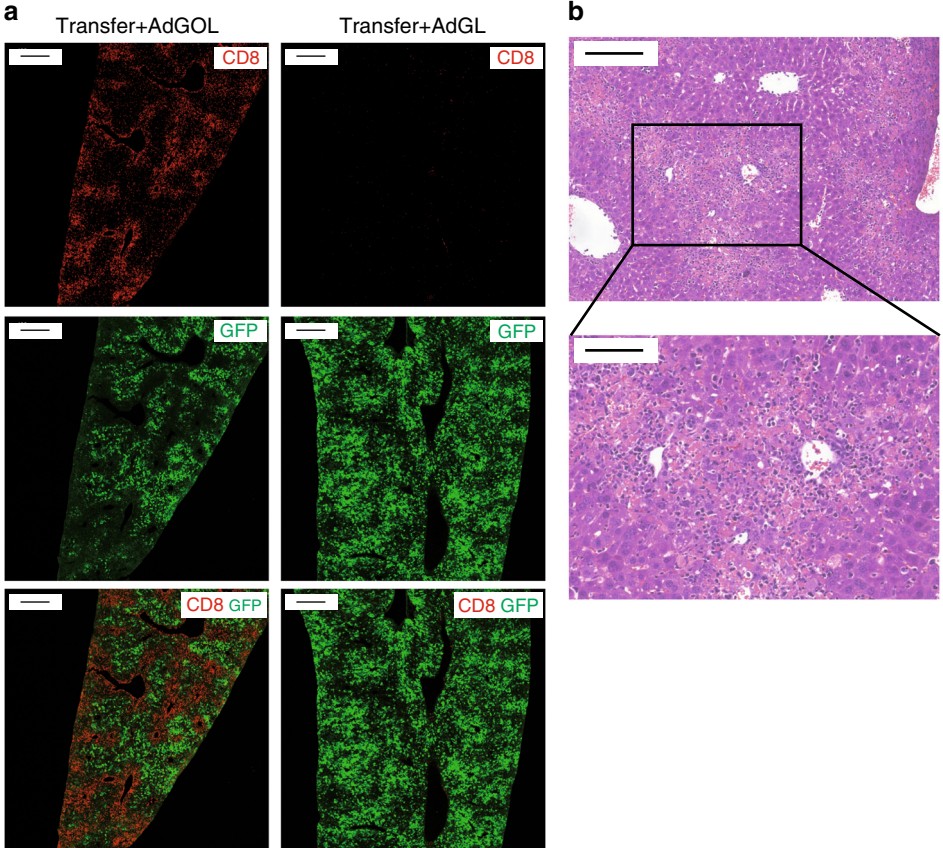

**Fig. 2** Antigen-specific CD8 T cells infiltrate in periportal areas. **a** Immunfluorescent images showing liver sections on day 3 after transfer of CD44⁻OT-I cells and infection with $1 \times 10^9$ PFU AdGOL (left row) or $1 \times 10^9$ PFU AdGL (right row). **b** H&E staining of liver section on day 3 after transfer of CD44⁻OT-I cells and infection with $1 \times 10^9$ PFU AdGOL. Scale bars: **a**: 400 μm, **b**: 200 μm/100 μm. Data are representative three independent experiments ($n = 10$)

remained in periportal areas and did not localize to infected hepatocytes in the perivenous areas (Fig. 2a). In hematoxylin and eosin (H&E) staining of liver tissue, we confirmed the massive immune cell accumulation around periportal areas accompanied by confluent areas of damaged hepatocytes that bridged between periportal areas (Fig. 2b).

**T cell effector mechanism during acute liver failure**. Having established that CD8 T cells infiltrate the liver in periportal areas, we sought to identify the effector function of CD8 T cells underlying the observed liver failure. To this end, we first screened for cytokines in the serum of mice on day 3 after AdGOL or AdGL infection and OT-I T cell transfer. We detected highly elevated, systemic levels of IFN-γ, TNF, interleukin (IL)-6 and IL-10 when CD8 T cells encountered their antigen in AdGOL- but not AdGL-infected animals (Fig. 3a). Initially, we focussed on IFN-γ as a T cell effector cytokine since IFN-γ was also found to be expressed during fulminant Hepatitis B[20]. However, Ifnγr⁻/⁻ mice showed a similar increase in serum ALT levels and loss of body weight as their wt counterparts and went on to develop fulminant hepatitis (Fig. 3b, c), arguing against a critical role for IFN-γ in the pathogenesis of liver failure. IL-6 can be produced by a variety of cells including monocytes, B cells, and CD4 T cells in response to microbial compounds[24]. Therefore, we speculated that T cell-induced hepatitis may involve microbial translocation from the gut, which is known to aggravate liver damage[25]. However, mice treated with antibiotics showed a similar extent of liver damage and loss of body weight as control animals

(Fig. 3d). Microbial translocation is therefore unlikely to contribute to the pathogenesis of fulminant viral hepatitis in this model. Next we addressed the role of TNF, a cytokine that is produced by both lymphocytes and myeloid cells, as well as the death ligand receptor pair FAS/FASL. Both molecules induce cell death and have been shown to play important roles in the pathogenesis of liver and gastrointestinal diseases[26,27]. However, in our model neither TNF blockade nor FASL blockade (Fig. 3f, g) changed the cause of fulminant viral hepatitis. Finally, we focused our attention on perforin-1, a protein that is stored in secretory vesicles with pro-apoptotic serine proteases (granzymes) of cytotoxic T cells[28]. Once an immune synapse forms, perforin and granzyme B are cosecreted into the immune synapse, where perforin forms transmembrane channels in the target cell plasma membrane, thus allowing the passive diffusion of granzyme B into the cytosol and inducing rapid target cell death[29,30]. To test the requirement for perforin-1 to induce hepatocyte death in our model, we transferred Prf1⁻/⁻ or wt OT-I T cells into AdGOL-infected wt recipients. Mice that received Prf1⁻/⁻ OT-I T cells did not have increased ALT levels and were fully protected from fulminant hepatitis (Fig. 3h). Despite an intial loss of body weight on day 2 similar to mice that received wt OT-I T cells, they regained weight by day 3 post infection and survived the infection (Fig. 3i, j). This early transient weight loss is probably independent from T cells because ALT elevation from antigen-specific immunity occurred only from day 2 onward. In summary, we show that fulminant hepatitis depends on a nonredundant function of perforin-1 in effector CD8 T cells.

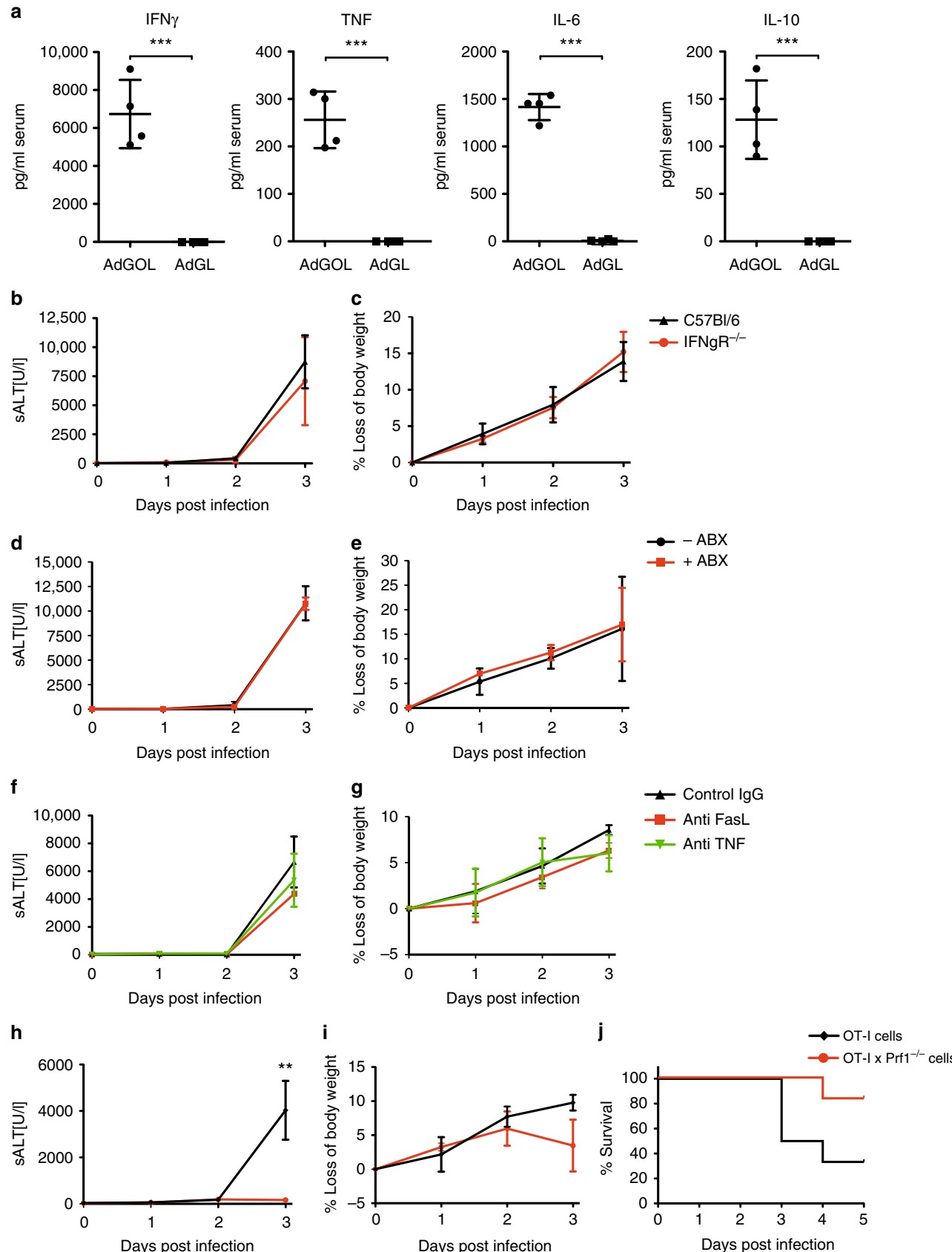

**Pathophysiology of acute T cell-mediated liver failure**. Having established the critical effector function of antigen-specific CD8 T cells, we next aimed to define parameters that would allow us to stratify mice that develop fulminant hepatitis versus those that recover. To this end, we used a model of TNF-mediated liver damage in the context of adenoviral infection[31]. In this approach, we infected mice with AdGOL and on day 2 post infection injected TNF i.v. (Fig. 4a). This model provides a high level of

**Fig. 3** Perforin-1 is critical for T cell-mediated acute liver failure. **a** Serum levels of IFN-γ, TNF, and IL-6 on day 3 after transfer of CD44⁻OT-I cells and infection with 1×10⁹ PFU AdGOL or 1×10⁹ PFU AdGL. **b, d, f, h** ALT activity in the serum and **c, e, g, i** percentage of loss of body weight in *ifnγr*⁻/⁻ mice (**b, c**), wt mice with or without antibiotics treatment starting at day −7 (**d, e**), wt mice that were treated anti-FasL, anti-TNF, or control antibody on day 2 post infection (**f, g**) or wt mice that received wt or *Prf1*⁻/⁻ OT-I T cells (**h, i**). **j** Survival curves of wt mice that received wt or *Prf1*⁻/⁻ OT-I T cells. Data are representative of two (**b–g**) or three independent experiments (*n* ≥ 3 (**d–i**) or *n* ≥ 4 (**a–c, j**)). Error bars indicate the mean ± SD. Comparison between groups was calculated using unpaired Student's *t* test (**a–i**) or log-rank test (**j**). ABX antibiotics \*\**p* < 0.01; \*\*\**p* < 0.001

control as the extent of liver damage can be fine-tuned by titrating the amount of applied TNF and/or the infectious dose of virus. Injection of 0.4 μg TNF led to similar ALT levels as seen with a OT-I T cell-mediated hepatitis and half of the infectious dose of AdGOL. Notably, despite high ALT levels, TNF-treated animals never developed fulminant hepatitis or succumbed to death (Fig. 4b). As expected, this result demonstrated that serum ALT levels correlated with the level of hepatocyte death; however, this parameter alone was not a good predictor of disease progression, which indicated that an additional factor may be critical to cause liver failure. Our observation that a substantial number of infected hepatocytes remained intact during viral hepatitis supports this notion (see Fig. 2a). Given the local accumulation of CD8 T cells in periportal areas of the liver, we suspected a possible involvement of the liver vasculature. First, we investigated the integrity of the liver sinusoidal vasculature by i.v. injecting Evans Blue. Based on the well-established function of TNF to regulate vascular permeability[32], our TNF-mediated hepatitis model served as a positive control for vascular leakage, which should lead to leakage of Evans Blue from the circulation and staining of parenchymal liver tissue. Indeed, we detected Evans Blue staining of the liver tissue in virus-infected mice that received TNF yet not in infected phosphate-buffered saline-treated animals (Fig. 4c). In AdGOL-infected mice that received OT-I T cells, we observed large areas with Evans Blue staining that bridged between adjacent portal areas (Fig. 4c), indicating a substantial loss of endothelial integrity after T cell-mediated hepatitis in a localized area. To provide further evidence for this notion, we stained liver tissue sections from the T cell- and cytokine-mediated hepatitis model by i.v. application of a CD31 antibody, in order to visualize the vascular tree/network. In control and TNF-treated animals, we observed a continuous vascular network (Fig. 4d). As expected, the vessels in such animals were slightly dilated as compared to control conditions, but the liver sinusoidal network was fully conserved (Fig. 4d). In contrast, the CD31 staining of the sinusoidal network in mice with CD8 T cell-mediated fulminant hepatitis was discontinuous and partly absent in periportal areas (Fig. 4d). Quantitative analysis of the vessel surface confirmed this visual impression (Fig. 4e, f). Finally, to link the observed alterations of the vasculature to perforin-mediated effector function of OT-I T cells, we compared the vascular leakage in mice that received wt OT-I T cells, with or without blockade of FASL or TNF with animals that received *Prf1*⁻/⁻ OT-I T cells. Only in AdGOL-infected mice that received *Prf1*⁻/⁻ OT-I T cells, no Evans Blue staining was observed (Fig. 4g). In summary, this data indicates that OT-I T cells directly or indirectly altered the liver sinusoidal endothelial function and that this may contribute to acute, fulminant viral hepatitis.

Next, we aimed to directly visualize the sinusoidal liver perfusion during acute, fulminant viral hepatitis. To this end, we injected Evans Blue in anesthetized mice and imaged the organ-wide in vivo distribution of the dye. Before injection, we noted that the livers from mice with a CD8 T cell-mediated fulminant viral hepatitis had a lighter appearance as compared to control livers indicative for a disturbed vascular perfusion (Fig. 5a). After Evans Blue injection, the livers of control mice

were quickly and homogenously stained (Fig. 5a, Supplementary Movies 1–3). In contrast, the livers developing fulminant hepatitis showed a delayed and patchy staining with Evans Blue, providing direct evidence for an irregular and severely impaired hepatic blood perfusion (Fig. 5a, Supplementary Movie 4). In order to directly visualize the blood flow in the liver in vivo, we injected fluorochrome-labeled anti-platelet antibodies and imaged the blood flow in hepatic sinusoids by in vivo microscopy. In control mice, we observed a continuous blood flow throughout the entire sinusoidal network (Fig. 5b, Supplementary Movie 5). In contrast, in mice with fulminant viral hepatitis, we observed a discontinuous and irregular blood flow through liver sinusoids (Fig. 5b, Supplementary Movie 6). We conclude from these data that the liver sinusoidal perfusion was severely impaired during fulminant viral hepatitis.

**Critical role of LSECs.** Next we investigated whether the loss of LSEC function and consecutive impairment of sinusoidal perfusion was a direct result of the CD8 T cell-mediated attack on liver endothelium. To determine the relevance of antigen presentation by different liver cell populations, we generated different bone marrow chimeric animals with a restriction of H2-Kᵇ-mediated antigen presentation on myeloid and endothelial cells (Tie2-Kᵇ → Tie2-Kᵇ), on myeloid cells and hepatocytes (Tie2-Kᵇ → Crp-Kᵇ), or on myeloid cells only (Tie2-Kᵇ → DBA/2) (Fig. 6a). We transferred OT-I cells into those various recipients, infected them with AdGOL, and analyzed ALT levels and body weight over time. Only mice with H2-Kᵇ expression on myeloid and LSECs developed severe viral hepatitis and loss of body weight of around 10% (Fig. 6b, c). In contrast, mice that expressed H2-Kᵇ on myeloid cells and hepatocytes showed only a mild increase in ALT levels and loss of body weight of about 5%. Importantly, OT-I cell abundance in the spleen of all the experimental groups was similar arguing that the number of antigen-specific CD8 T cells did not account for the observed differences in disease severity (Fig. 6d). Finally, to link our functional data to the observed alterations of the LSEC network, we stained the livers from the various experimental groups with CD31 and visualized the infiltrating CD8 T cells. In mice with H2-Kᵇ-restriction on myeloid cells, the LSEC vascular network was fully intact and CD8 T cells were located within liver sinusoids (Fig. 6e, f). In mice with H2-Kᵇ expression on myeloid cells and hepatocytes, we observed infiltration of CD8 T cells into the liver parenchyma; however, the CD31 staining was continuous arguing for preserved integrity of the sinusoidal vasculature. In contrast, in mice with H2-Kᵇ expression on myeloid and LSECs the CD31 staining was discontinuous, particularly in areas of CD8 T cell accumulation in the periportal zones (Fig. 6e, f).

We have previously shown that our recombinant adenoviruses selectively infect hepatocytes but not LSECs[33]. However, we were concerned that a very low level of viral antigen expression within LSECs may be the basis for T cell-mediated attack. To rule out this possibility, we generated a recombinant adenovirus that expresses OVA under the hepatocyte-specific TTR (amyloid precursor transthyretin) promoter. As expected, the overall level of antigen expression in vivo as compared to CMV promoter-

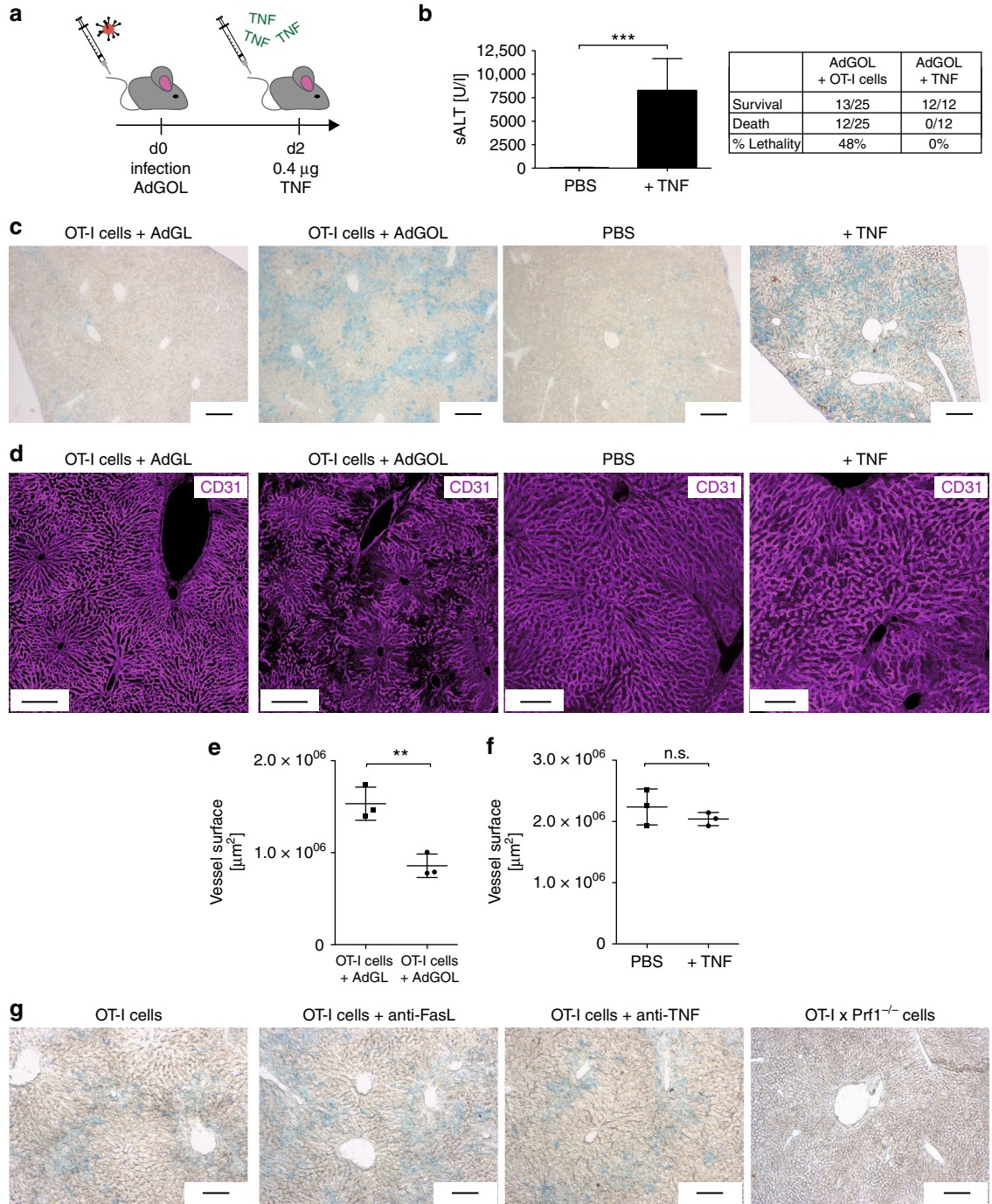

**Fig. 4** Disturbed endothelial integrity of liver sinusoids is an essential component of fatality. **a** Schematic showing the approach to induce transient, TNF-mediated hepatitis after adenoviral infection. **b** Serum ALT activity of mice infected with $2 \times 10^9$ PFU AdGOL and injected with 400 ng TNF i.v. on day 2 post infection. Samples were taken 4 h after injection of TNF. **c** Light microscopic images of the livers of mice with acute liver failure (T cell-mediated) or with transient hepatitis (cytokine-mediated) showing Evans Blue leakage. Evans Blue dye was injected i.v. on day 3 post OT-I cell transfer and infection with AdGOL or 4 h post TNF injection. **d**–**f** Immunofluorescence images showing CD31 staining (**d**) and quantification of vessel surfaces (**e**, **f**) of the livers with T cell- or cytokine-mediated hepatitis. **g** Light microscopic images of the livers of wt mice adoptively transferred with wt or $Prf1^{-/-}$ OT-I cells and infection with AdGOL on day 0. Some mice were injected with FasL- or TNF-blocking antibodies on day 2 and Evans Blue dye was injected on day 3. Scale bars: 200 μm (**c**, **g**) or 100 μm (**d**). Data are representative of two (**b**) or three independent experiments ($n = 5$ (**b**) or $n = 3$). Error bars indicate the mean ± SD. Comparison between groups was calculated using unpaired Student's $t$ test. n.s., not significant; **$p < 0.01$; ***$p < 0.001$

driven antigens was significantly reduced (about 15-fold) (Supplementary Fig. 3A). Conclusively, the ALT serum levels were about five-fold reduced in the T cell-mediated hepatitis model after adenovirus infection with the TTR-driven as compared to the CMV-driven construct (Supplementary Fig. 3B). Importantly, when we analyzed the tissue section from animals infected with the TTR-driven construct, we still detected periportal infiltration of CD8 T cells that colocalized with a disrupted vascular network as revealed by CD31 staining (Supplementary Fig. 3C, D). Since LSEC do not express OVA after infection with the TTR-driven adenovirus, this data provide strong evidence for cross-presentation of viral antigen from

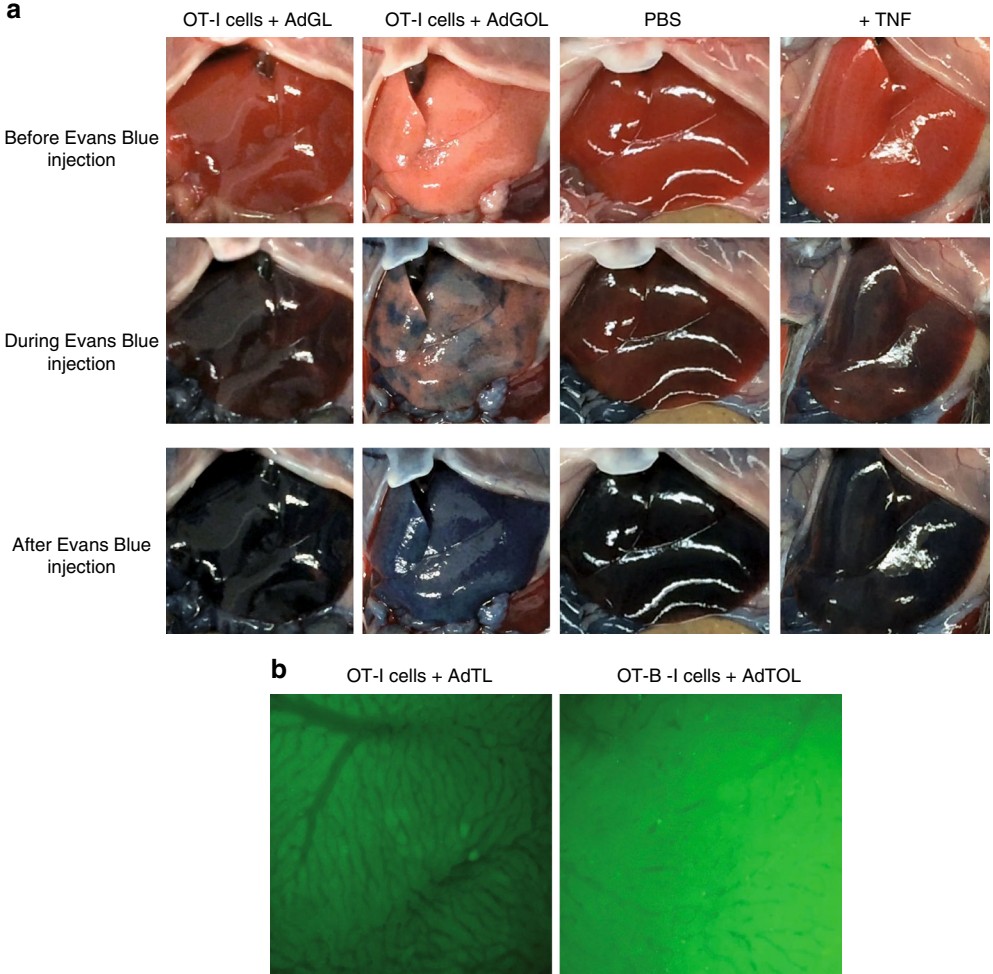

**Fig. 5** Vascular perfusion is severely impaired in the failing livers. **a** Serial images showing the distribution of Evans Blue dye after intravenous injection in mice with T-cell (OT-I) or cytokine (TNF) mediated hepatitis. See also Supplementary Movies 1–4. **b** Microvascular perfusion of the livers of control mice (AdTL) or mice with T cell-mediated hepatitis (AdGOL) on day 3 post infection. See also Supplementary Movies 5 and 6. Data are representative of two independent experiments ($n = 4$)

infected hepatocytes by LSECs, which then become targets for CTL-mediated attack. To formally test that LSECs activate CD8 T cells in an antigen-dependent manner, we isolated this cell population after infection and coincubated them with CFSE-labeled OT-I T cells. Ex vivo, LSECs from AdGOL-infected mice induced strong proliferation of OT-I cells, in contrast to LSEC isolated from AdGL-infected animals (Supplementary Fig. 3E). In summary, we conclude that antigen cross-presentation by LSECs causes T cell-mediated elimination that is both required and sufficient to cause fulminant hepatitis.

**Therapy of CD8 T cell-mediated acute liver failure**. Finally, based on our results we aimed to test a therapeutic approach for fulminant hepatitis. We have recently developed a new class of small molecule perforin inhibitors based on benzenesulphona-mide chemistry that are suitable for use in vivo[34,35]. The compounds potently inhibit human and murine perforin, but have no effect on TNF-mediated cell death in vitro (Supplementary Fig. 4). To test whether pharmacological inhibition of perforin-1 in vivo can influence the course of T cell-mediated hepatitis, we treated AdGOL-infected mice that received OT-I T cells with the perforin-1 inhibitor SN34960, starting on day 1 after infection. We found that pharmacological perforin-1 inhibition lead to a significant reduction of ALT levels (Fig. 7a). Loss of body weight

was slightly more pronounced during the early phase of therapy (Fig. 7b), the cause of which we are currently exploring. Strikingly, mice that were treated with the perforin-1 inhibitor were fully protected from lethality in contrast to solvent-treated animals (Fig. 7c). Conclusively, when analyzing liver tissue by immunofluorescence (IF), we found that liver sinusoids remained intact in mice treated with the perforin-1 inhibitor (Fig. 7d). Also, the abundance of accumulating CD8 T cells was reduced in perforin-1 inhibitor-treated animals compared to mock-treated mice, yet this reduction in hepatic CD8 T cells was not accompanied by a reduction in the numbers of infected GFP-expressing hepatocytes (Fig. 7d). This observation argues for a feed-forward mechanism in which T cell-mediated elimination of antigen-presenting cells lining liver sinusoids leads to further recruitment of antigen-specific CD8 T cells from the vasculature into the liver parenchyma, where further infected cells can be eliminated.

## Discussion

Here we provide evidence for a new mechanistic understanding of fulminant viral hepatitis: besides direct targeting of virus-infected hepatocytes, critical liver damage is induced by CD8 T cell effector function against non-infected LSECs that cross-present antigens released from infected hepatocytes. The severe sinusoidal perfusion deficits of the liver resulting from antigen-specific CD8

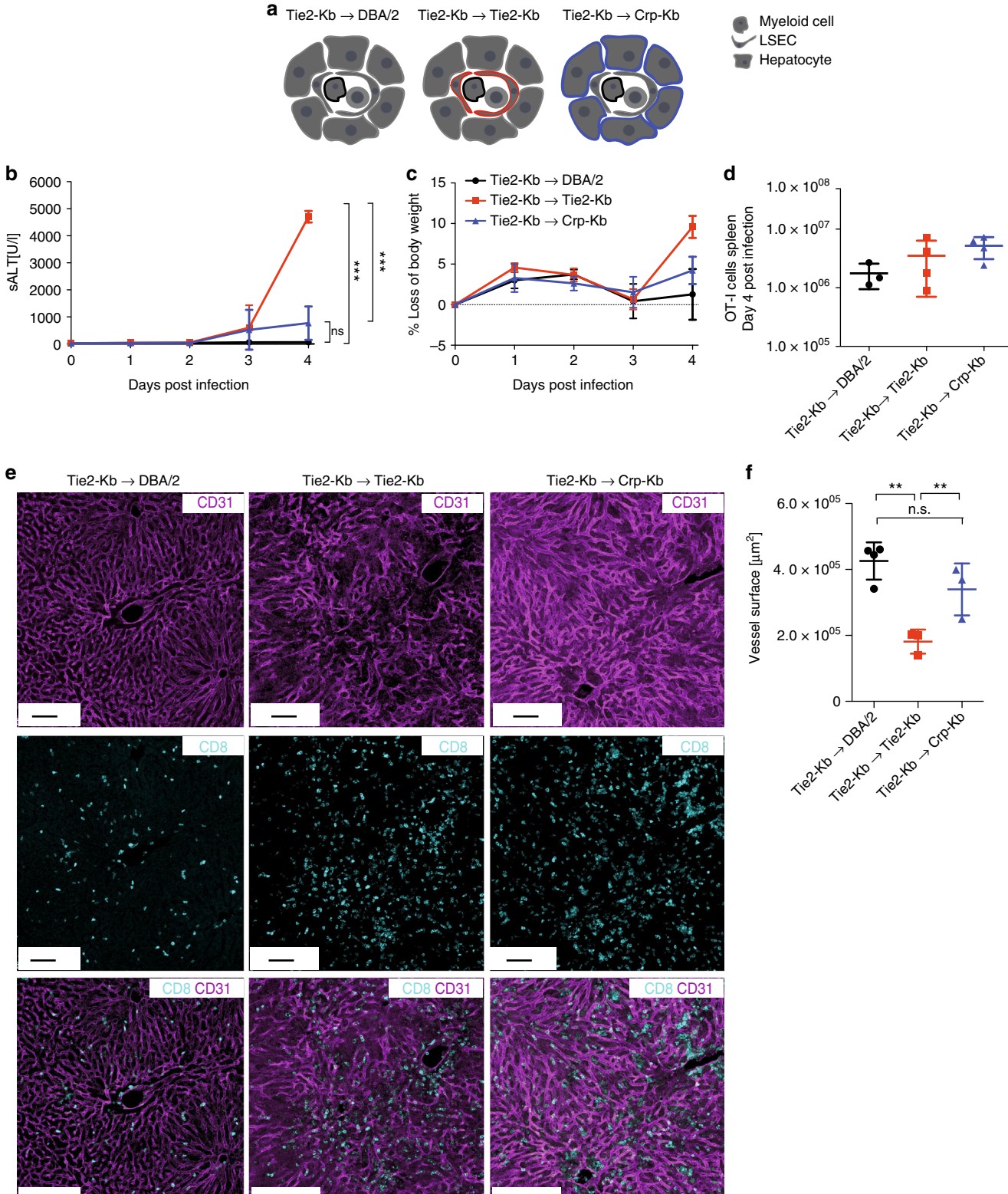

**Fig. 6** Antigen presentation on LSECs in the liver is sufficient to drive liver failure. **a** Schematic of various bm-chimeric animals that have a Kb-restricted antigen presentation on myeloid cells only (Tie2-Kb→DBA/2, black), myeloid and endothelial cells (Tie2-Kb→Tie2-Kb, red), or myeloid cells and hepatocytes (Tie2-Kb→Crp-Kb, blue). **b**–**d** ALT serum levels (**b**), percentage of loss of body weight (**c**) and absolute numbers of OT-I cells recovered from the spleen of the various bm-chimeric animals (**d**). **e** Immunofluorescence images showing CD31 and CD8 staining of the livers of various bm-chimeric mice on day 4 post infection. **f** Quantification of vessel surface of the various bm-chimeric mice. Data are representative of three independent experiments (n ≥ 3). Scale bars: 80 μm. Error bars indicate the mean ± SD. Comparison between groups was calculated using one-way ANOVA with Turkey's multiple comparison post test. n.s., not significant; *p ≤ 0.05; **p < 0.01; ***p < 0.001

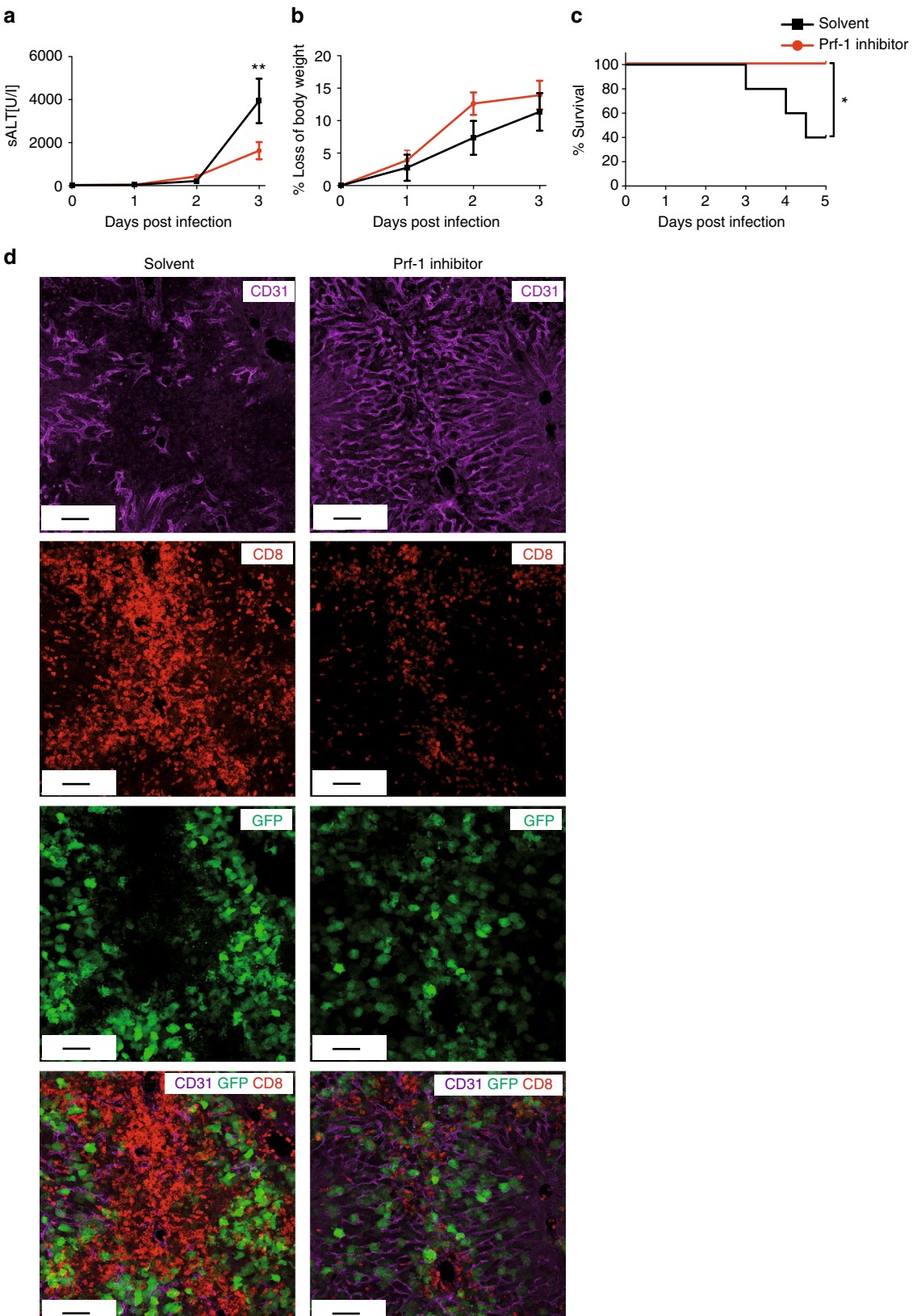

**Fig. 7** Therapeutic treatment with perforin-1 inhibitor ameliorates the liver damage. **a–c** Serum ALT activity (**a**), percentage of loss of body weight (**b**), and survival curves (**c**) of wt mice with T cell-mediated hepatitis that received perforin-1 inhibitor or solvent treatment starting at 24 h post infection. **d** Immunofluorescence images of the mouse livers treated as in **a–c** showing CD31, CD8, and virally expressed GFP. Data are representative of two independent experiments ($n \geq 5$). Scale bars: 80 μm. Error bars indicate the mean ± SD. Comparison between groups was calculated using unpaired Student's $t$ test. *$p \leq 0.05$; **$p < 0.01$

T cell effector function against LSECs required perforin-1 as non-redundant effector mechanism and triggered liver failure. Using a novel pharmacological perforin-1 inhibitor, we demonstrate that blocking perforin-1 function rescues mice with viral hepatitis from fulminant liver failure, which may provide a salvage treatment strategy for patients with fulminant viral hepatitis.

We have established a preclinical murine model for fulminant hepatitis using recombinant adenoviruses that target hepatocytes[36] similar to hepatotropic hepatitis viruses and the high-affinity TCR OT-I system that reflects the high-affinity virus-specific CD8 T cells found during acute viral hepatitis[37]. We would like to point out that this is not a model for Hepatitis B infection; however, similar mechanisms as revealed here may be at play in patients with Hepatitis B that deteriorate into liver failure based on T cell-mediated attack. Notably, during systemic LCMV infection in mice, endothelial cells are also targets of a perforin-dependent CTL-mediated attack. This T cell-mediated elimination of endothelial cells is inhibited by programmed death ligand 1 (PD-L1) interactions but leads to rapid death of mice that are genetically deficient of this check-point molecule[38].This argues that this pathophysiological mechanism may be a common pattern in deteriorating viral infections in which specific regulatory mechansims are annuled. In this regard, the function and pathophysiological role of LSECs in the context of immune-mediated liver pathology is largely understudied. Currently, it is unknown under which inflammatory conditions human LSECs take up extracellular antigen, process, and present it via major histocompatibility complex class I (cross-presentation) or how they regulate PD-L1 expression.

This study mirrors a fulminant viral hepatitis in patients regarding the observed massive hepatic accumulation of lymphocytes, while granulocytes did not have an active role in pathogenesis of fulminant hepatitis. Our model therefore provides a significant advantage over existing models of toxic liver failure[39] or liver failure from antigen-independent stimulation of immune cell effector functions[40]. This allowed us to dissect the cellular and molecular mechanisms underlying fulminant viral hepatitis. Also it appears that a combination of high numbers of virus-infected hepatocytes together with high numbers of antigen-specific CD8 T cells in the periportal area need to coincide in order to trigger immunopathology and liver failure. This fatal combination may occur during acute fulminant hepatitis, during the immune reconstitution inflammatory syndrome in HIV/HBV co-infected patients[41,42] or in patients undergoing rituximab treatment.

Contrary to the belief that killing of infected hepatocytes is exclusively responsible for liver damage, we found that CD8 T cell effector function against infected hepatocytes alone was not sufficient to cause an acute liver failure. Rather, CD8 T cell targeting of LSEC cross-presenting antigens secreted or released from killed infected hepatocytes was also required to trigger liver failure. In the presence of antigen-experienced effector T cells, the capacity of LSECs to induce tolerance is limited[43]. LSEC cross-presentation of hepatocyte-derived antigens to low numbers of antigen-specific CD8 T cells initiates TNF production that contributes to 50% of total antiviral CD8 T cell immunity via pro-apoptotic signaling through the TNF receptor selectively in infected hepatocytes[31]. This TNF-dependent non-canonical CD8 T cell effector function is not followed by liver failure despite severe liver damage[31]. However, in the presence of very high numbers of CD8 T cells, this beneficial antiviral immune defence through cross-presenting LSECs turns into a life-threatening damage-inflicting process. Antigen-specific CD8 T cell attacking cross-presenting LSECs caused severe liver perfusion defects that consequently led to secondary hepatocyte death and ultimately liver failure. The preferential accumulation of CD8 T cells in the periportal area that we observed may result from trapping of blood-borne antigen-specific CD8 T cells passing through liver sinusoids by cross-presenting LSECs[44]. Since blood-borne CD8 T cells enter the sinusoidal circulation in the periportal region, where portal vein branches merge with arterial blood to give rise to low pressure perfusion in liver sinusoids, periportal accumulation may simply result from immediate trapping of antigen-specific CD8 T cells once they enter the liver sinusoids and meet cross-presenting LSECs. Of note, antigen-specific CD8 T cells remained in the periportal areas although virus-infected hepatocytes were present further downstream of sinusoidal blood flow in perivenous areas, which is consistent with local antigen presentation that is known to provide a migration stop signal for T cells[45]. We did not find evidence for sinusoidal thrombosis as cause for sinusoidal perfusion failure (Supplementary Movie 6). Rather, we found that perforin-1-dependent killing of cross-presenting LSECs is the predominant and non-redundant effector mechanism of CD8 T cells during fulminant viral hepatitis. In the absence of perforin-1 expression or pharmacological inhibition, CD8 T cell accumulation in periportal areas was reduced, sinusoidal perfusion was not impaired, and livers did not fail, indicating that periportal immobilization of CD8 T cells following LSEC killing may have caused the sinusoidal perfusion collaps. This identifies perforin-dependent killing of cross-presenting LSECs by CD8 T cells as lethal incident during antiviral immune responses leading to fulminant viral hepatitis, when high numbers of CD8 T cells are present. Having identified this pathophysiological mechanism, we were able to develop a specific therapy with a new class of perforin-1 inhibitor that was highly effective in a preclinical animal model. Future studies are required to validate whether this pathophysiological mechanism is also critical in human disease. If so, pharmacological inhibiton of perforin-1 may be helpful in patients with a high risk of developing fulminant viral hepatitis to prevent further disease aggravation and transition into liver failure.

## Methods

**Mice**. For all experiments, age- and sex-matched animals were used and littermate controls whenever applicable. C57Bl/6 mice, OT-I mice, $Prf$-$1^{-/-}$ mice, and DBA/2 mice were purchased at Jackson or Janvier Labs and kept in in-house facilities. Tie2-K$^b$ mice and Crp-K$^b$ mice were bred in in-house facilities. $Prf$-$1^{-/-}$ mice were bred to OT-I mice. $Ifnyr$ KO mice were kindly provided by I. Förster. Bone marrow chimeric mice were generated as previosly reported[46]. Animal experiments were approved by the Animal Care Commission of the state of North Rhine-Westphalia.

**Viruses**. E1- and E3-deleted adenoviral vectors expressing eGFP or tdTomato and OVA and CBG99-Luciferase (AdGOL, AdTOL), eGFP, and CBG99-Luciferase (AdGL) or tdTomato and CBG99-Luciferase (AdTL) were generated and used as previously described[31]. In Ad(CMV)GOL, gene expression was driven by the CMV promoter, and in Ad(TTR)GOL the CMV promoter was replaced by the murine TTR promoter for hepatocyte-restricted gene expression. For generation of Ad-GOL-Gp$_{33}$, the DNA sequence encoding for amino acids QLE**SIINFEKL**TEW in Ad-GOL was replaced by a DNA sequence encoding for amino acids TSI**KA-VYNFATC**GVF, the immunodominant peptide (in bold) recognized by CD8 T cells from the TCR transgenic mouse line P14.

**Generation of T cell memory**. A total of $1 \times 10^4$ CD44 negative, OT-I cells were isolated via MACS purification (Miltenyi) and transferred into C57Bl/6 mice i.v. Mice were immunized with 0.5 mg OVA, 0.05 mg poly I:C, and 0.05 mg anti-CD40 antibody intraperitoneally (i.p.) on the same day.

**Hepatitis induction**. Naive mice were transferred with $7 \times 10^6$ naive (CD44 negative) OT-I cells and infected with $1 \times 10^9$ plaque-forming unit (PFU) adenovirus on day 0. Memory mice were infected with $1 \times 10^9$ PFU adenovirus 30 days after OT-I cell transfer and/or immunization (see also "Generation of T cell memory"). For the induction of a cytokine-mediated hepatitis, mice were infected with $2 \times 10^9$ PFU AdGOL i.v. and injected with 0.4 µg TNF i.v. on day 2 after infection. ALT levels in the serum were measured with the Reflotron®plus system (Roche).

**Histology and imaging analysis**. For IF stainings, paraformaldehyde (PFA)-lysine-periodate (PLP)-fixed, frozen tissues were cut, stained, mounted, and acquired on a 710 confocal microscope (Carl Zeiss Microimaging). The following antibodies were used: CD8-PE (clone 5H10, Invitrogen, 1:200), GFP-AF488 (polyclonal, Life Technologies, 1:500), and CD31-AF647 (clone Mec13.3, Biolegend 1:100). Total vessel surface was quantified using the surface tool of Imaris software (Bitplane). For H&E stainings, the livers were fixed in 4% PFA and embedded in paraffin. H&E stainings were performed on deparaffinized sections with Eosin and Mayer's Haemalaun following a standard protocol.

**Assessment of vascular perfusion of the liver**. Evans Blue dye 2 mg was injected i.v. under general anesthesia. The liver was explanted 1 min after injection, fixed in PLP buffer, and dehydrated in 30% sucrose for 1 day each and subsequently cut, mounted, and imaged with a IX71 light microscope (Olympus). To assess CD31 staining of the liver vasculature, 10 µg of fluorescent anti-CD31 antibody (AF647-labeled, clone Mec13.3, Biolegend) was injected i.v. 15 min before explantation of the liver. For intravital microscopy, mice were anesthetized and the liver was carefully exposed and touched to a glass slide. Two micrograms of fluorescent anti-GP1bβ antibody (Emfret Analytics) was injected i.v. for visualization of platelets and the blood flow and video acquisition was carried out with epifluorescent light.

**Inhibitors and blocking antibodies**. In all, 500 µg anti-FasL (antiCD178, Biolegend) or 500 µg anti-TNF antibody (Infliximab, Janssen Biotech) were injected i.p. on day 2 post infection.

Mice were injected with 150 mg/kg of the perforin inhibitor SN34960 in solvent (20% (2-hydroxypropyl)-β-cyclodextrin) i.p. at 12, 24, and 36 h post infection.

**Flow cytometry**. For fluorescence-activated cell sorting (FACS) analysis of cells from the blood, cells were stained with anti-CD8-PE antibody and H2-K$^b$/SIINFEKL-Dextramer-APC (Immudex, 1:50). For analysis of cells from the spleens, the tissue was dissociated and cells were stained with anti-CD45.1-APC (clone A20, Biolegend, 1:100) and anti-CD8-PE (clone 5H10, Invitrogen, 1:200). Live/dead™ Fixable Near-infrared Dead Cell Stain Kit (ThermoFisher) was used to exclude dead cells. CountBright™ absolute counting beads (ThermoFisher) were added for the calculation of absolute cell numbers per organ. Data were acquired with FACSCanto™ II or LSRFortessa™ flow cytometers (BD Biosciences) and analyzed with the FlowJo Software (Tree Star Inc.). For FACS-based cell sorting, we used FACSAria™ III.

**Statistics**. Statistical significances were calculated with an unpaired Student's $t$ test, a one-way analysis of variance with Tukey's multiple comparison post test or a log-rank test as indicated. Statistical significances are marked as *$p \leq 0.05$; **$p < 0.01$; ***$p < 0.001$.

**Reporting summary**. Further information on research design is available in the Nature Research Reporting Summary linked to this article.

## Data availability
The data that support the findings of this study are available from the corresponding author upon reasonable request.

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

## Acknowledgements

We would like to thank C. Boerner, S. Ebbinghaus, S. Rathmann, and K. Dumler for technical assistance. This research was supported by the german research foundation DFG (SFB 704, SFB 670, and TRR 179). W.K., Z.A., and C.K. are members of the DFG Excellence Cluster ImmunoSensation in Bonn, Germany. W.K. is supported by NRW-Rückkehrerprogramm of the German state of Northrhine-Westfalia. D.W. was supported from BONFOR. J.T. was supported by DFG (SFB TRR57 P18), European Union's Horizon 2020 research, the innovation programme (No. 668031), and the Cellex Foundation. P.K. was supported by the German Center for Infection Research, Munich Site.

## Author contributions

M.W. and S.E. planned and performed the experiments and analyzed the data. W.K., P.K. and D.W. conceptualized the study and analyzed data. A.J.D. helped with histology. B.N. and D.Z. provided critical reagents. W.K., P.K. and D.W. provided research funds. J.A.T., K.H.G. and J.A.S. developed and provided the perforin-1 inhibitor, planned experiments, and analyzed data. J.T., Z.A., C.K., H.A., M.A. and K.M. were involved in study design and data analyses. P.K. and W.K. wrote the manuscript with input from all authors. M.W. designed the figure graphics.

## Additional information

**Competing interests:** The authors declare no competing interests.

