## [Peer Review File · Nature Communications]

Reviewers' comments:

Reviewer #1 (Remarks to the Author):

The manuscript entitled "Perforin inhibition can prevent acute CD8 T cell-mediated liver failure during viral hepatitis" by Welz et al. developed a new mouse model for an acute fulminant viral hepatitis. They found that perforin-mediated killing is a critical function of Ag-specific CD8 T cells during fulminant hepatitis and a perforin-1 inhibitor can rescue CD8 T cell-mediated liver damage. Overall, the experiments are in general well controlled and clearly described. The conclusions are supported by the experimental evidence presented. However, the authors should address the following criticisms:

1. The authors used OT-1 cells and ovalbumin-expressing recombinant adenovirus systems to develop a new mouse model with fulminant viral hepatitis. However, since OT-I TCR would have extremely higher TCR affinity than TCR of virus-specific T cells, there is a possibility that OT-I system used in this mouse model does not accurately reflect the condition of CD8 T cells during acute fulminant viral hepatitis. Therefore, the experiments need to be performed using a model other than the OT-I / ovalbumin-expressing Ad virus system.
2. The authors should experimentally confirm that perforin-mediated liver damage by Ag-specific CD8 T cells during fulminant viral hepatitis is indeed a major cause of liver failure in fulminant viral hepatitis in human patients. The authors may characterize the virus-specific CD8 T cells using clinical samples from patients with fulminant viral hepatitis.
3. Figure 1F-G: Did the authors investigate the survival rate?
4. Figure 1 and 7: It would be nice if the authors can show the viral titers of challenged adenovirus in each group.

Reviewer #2 (Remarks to the Author):

Welz and colleagues present data on the role of perforin in fulminant viral hepatitis. They use a novel mouse model of a T-cell mediated viral hepatitis using a recombinant adenovirus expressing ovalbulmin and show that virus-specific endogenous memory T cells and even naïve T cells can induce fulminant liver injury after infection. They then explored the cause of the liver injury. Multiple cytokines were elevated in animals with fulminant hepatitis. To clarify their role, they show that IFN-gamma receptor KO mice, mice treated with antibiotics (bacterial translocation), anti-TNF and anti-FASL-treated mice had a similar fate to wt mice, suggesting a limited role in fulminant hepatitis. However, T cells lack perforin showed a much different phenotype with some weight loss but no ALT rise. They then used TNF to affect vascular integrity, showing that T-cell-mediated hepatitis led to a greater loss of vascular integrity, which was absent in mice with perf^{-/-} T cells. To clarify why LSEC were lost, they used bone marrow chimeric mice to limit Ag presentation and showed that only animals with Ag expression on myeloid cells and LSEC but not myeloid cells and hepatocytes or myeloid alone developed severe hepatitis and had loss of vascular integrity. To exclude the possibility of low-level infection of LSEC, they generated an adenovirus with ovalbumin under the TTR receptor and although this led to a less dramatic hepatitis, they still saw the same targeting of LSEC suggesting cross-presentation of Ag from hepatocytes to LSEC. Finally they used a novel pharmacologic inhibitor of perforin and showed that it limited the severity of the T-cell mediated hepatitis.

This is a very nicely performed and well-presented set of experiments. There are some minor considerations that merit further consideration.

1. The model is very useful for limiting the effects to a T cell-mediated fulminant hepatitis. In natural models of fulminant hepatitis (e.g. MHV infection), other mechanisms may be important as well. Although this model is useful to tease apart the effector mechanisms of a T-cell mediated hepatitis, it would be helpful to show at least some of the same findings in a more physiological hepatitis infection. A demonstration that the perforin inhibitor has an effect in MHV or any 'natural' fulminant hepatitis would make the findings much more compelling in terms of their potential utility.
2. Although it is implied that survival was better in the mice without fulminant hepatitis, survival curves are only shown in figure 1. It would be helpful to show these in figures 3 and most importantly in figure 7 as well.
3. Some further granularity on the effects in a few settings would be helpful. It is mentioned that the number of naïve T cells and the viral inoculum determined whether a severe or fulminant hepatitis occurred. It would be helpful to show these figures and ideally the difference in the pathology between in the 2 settings.
4. It would also be helpful to show a dose response curve for the pharmacological perforin inhibitor. Is there anything about the degree of perforin inhibition by this agent ie is the damage that is seen due to incomplete perforin inhibition or other mechanisms?
5. The authors expected that the number of virus-specific T cells pre-treatment would determine the outcome after infection (Fig 11) but conclude this is not the case based on the correlation between T cell number and ALT. In truth, the curve looks like there is a fairly good correlation that is largely affected by 1 or possibly 2 outliers. Certainly not a major issue, but I would suggest that they not dismiss this association so strongly.

Point-by-point response to the reviewer's questions:

We thank the reviewers for taking the time to assess our manuscript in detail and to provide their critical comments on our results. We strongly believe that pharmacological perforin inhibition has the potential to treat patients with T cell and NK cell mediated pathologies and in particular fulminant hepatitis. From a pathophysiological perspective, we believe our study makes an important advance: it demonstrates that the breakdown of tolerance mechanisms that normally protect endothelial cells from CTL-mediated attack can be the major contributor to lymphocyte mediated organ failure, rather than T cell killing of virus-infected parenchymal cells, as has been assumed until now.

Below, we have addressed the concerns of reviewers in a point-by-point response:

Reviewer 1:

1. The authors used OT-1 cells and ovalbumin-expressing recombinant adenovirus systems to develop a new mouse model with fulminant viral hepatitis. However, since OT-I TCR would have extremely higher TCR affinity than TCR of virus-specific T cells, there is a possibility that OT-I system used in this mouse model does not accurately reflect the condition of CD8 T cells during acute fulminant viral hepatitis. Therefore, the experiments need to be performed using a model other than the OT-I / ovalbumin-expressing Ad virus system.

Response: The reviewer raises concerns regarding the OT-I /ovalbumin system and expresses doubt as to whether this model can faithfully recapitulate the pathogenesis of an authentic viral hepatitis. We agree with this reviewer that OT-I T cells have a high affinity for their cognate peptide MHC complex (Kb/OVA₂₅₇). However, to our knowledge there is no example in the literature in which the results or conclusions obtained by the use OT-I T cells were not representative for the biology of CD8 T cells in general. Additionally, we would like to point out that antiviral CD8 T cells are also typically characterized by a high affinity towards viral antigens that are foreign antigens against which high-affinity TCRs are present in the entire T cell population. We further show that endogenous polyclonal CD8 T cells also cause a severe hepatitis and weight loss in mice in our model (Figure 1G/H).

However, we also understand the concerns raised by this reviewer. We therefore generated a new recombinant adenovirus that expresses a viral epitope derived from the glycoprotein (GP33) of murine LCMV (AdGP33). Importantly, GP33-specific TCR transgenic T cells (P14) cause a fulminant hepatitis and death of AdGP33 infected mice (Figure S2), similar to OT-I T cells in the context of Ad-GOL infection. Thus, fulminant viral T cell mediated hepatitis also develops when an authentic viral antigen is used as the immunogen in our recombinant adenovirus model.

2. The authors should experimentally confirm that perforin-mediated liver damage by

Ag-specific CD8 T cells during fulminant viral hepatitis is indeed a major cause of liver failure in fulminant viral hepatitis in human patients. The authors may characterize the virus-specific CD8 T cells using clinical samples from patients with fulminant viral hepatitis.

Response: We agree with the reviewer that analysing patient samples and translating our results and therapeutic approach to patients with fulminant hepatitis and other diseases in which Prf1 plays a critical role like GvHD would be very interesting. However, it is not possible to obtain liver tissue from patients with fulminant hepatitis, since liver biopsy is not generally indicated in this dire clinical situation. The suggestion therefore goes beyond the scope of the current manuscript. Also, in the final analysis, whether the remarkable protection of the perforin inhibitor-treated mice observed in our study can translate to humans can only be answered with a properly constituted clinical trial.

3. Figure 1F-G: Did the authors investigate the survival rate?

Response: In Figure 1F-G we analyzed the capacity of endogenous CD8 T cells to cause a severe hepatitis in our model. This is indeed the case, however the mice do not succumb, but recover after losing up to 15% of their body weight.

4. Figure 1 and 7: It would be nice if the authors can show the viral titers of challenged adenovirus in each group.

Response: The recombinant adenovirus used throughout this study lack E1/E3 and are therefore replication-deficient. They can infect hepatocytes and induce expression of the encoded transgenes, but do not replicate. We therefore always refer to the virus dose or the inoculation, which is directly correlated to the numbers of virus-infected hepatocytes. In Figure 2A (related to Figure 1) we show GFP expressing cells using AdGL vs AdGOL infection. Here, we clearly demonstrate that a large fraction of GFP expressing hepatocytes is eliminated in an antigen-specific fashion by antigen-specific CD8 T cells. However, a significant fraction of AdGOL-expressing hepatocytes remains. This led us to conclude that an additional mechanism besides direct elimination of infected hepatocytes is required for liver failure. Our further studies confirmed that this is due to prf1-mediated elimination of endothelial cells.

Regarding Figure 7 please see our comments to question 2 by Reviewer 2. We now provide new data demonstrating that pharmacological prf1 inhibition fully protects mice from lethality in our model.

Reviewer 2:

1. The model is very useful for limiting the effects to a T cell-mediated fulminant hepatitis. In natural models of fulminant hepatitis (e.g. MHV infection), other

mechanisms may be important as well. Although this model is useful to tease apart the effector mechanisms of a T-cell mediated hepatitis, it would be helpful to show at least some of the same findings in a more physiological hepatitis infection. A demonstration that the perforin inhibitor has an effect in MHV or any 'natural' fulminant hepatitis would make the findings much more compelling in terms of their potential utility.

Response: The reviewer raises an important issue and we are aware of the limitations of our model. However, to our knowledge this is currently the first and only preclinical model of a fulminant CD8 T cell mediated viral hepatitis in mice. Notably, CD8 T cells are typically associated with fulminant hepatitis in humans (PMID: 15566505), yet the mode of action explaining how virus-specific CD8 T cells trigger *fulminant* viral hepatitis has remained elusive.

Mouse hepatitis virus has often been used in the past to study principles of viral infection, but for many reasons it is not a good model of fulminant human hepatitis. Despite its name, mouse hepatitis virus, MHV is a coronavirus that besides the liver infects the CNS, all kind of epithelial cells, macrophages and also lymphocytes. Thus, it has a substantially different tropism compared to viruses typically causing fulminant viral hepatitis in humans, such as the hepatitis B virus. Given the above reasons, but in particular because MHV does not cause liver failure in mice, we did not consider MHV infection as a suitable model, but rather decided to employ a hepatotropic adenovirus for delivery of an immunogen. Mechanistically, replication-deficient recombinant adenoviruses have the further advantage that they allow to determine the minimal number of infected hepatocytes required for fulminant viral hepatitis to occur.

Following up on the question raised by Reviewer 1 regarding OVA and OT-I T cells as a model system, we generated a new recombinant Ad virus that expresses a viral epitope derived from the GP33 glycoprotein of murine LCMV (AdGP33). Using this AdGP33 and gp33-specific CD8 T cells we also observe fulminant viral hepatitis (Figure S2). Thus, we can fully recapitulate the data obtained with AdGOL and OT-I T cells. Also we would like to point out that PD-1 deficient animals succumb to lethal LCMV infection that is caused by prf1-mediated attack on endothelial cells (Frebel et al JEM 2012, PMID: 23230000), indicating the relevance of this mechanism during a natural infection.

2. Although it is implied that survival was better in the mice without fulminant hepatitis, survival curves are only shown in figure 1. It would be helpful to show these in figures 3 and most importantly in figure 7 as well.

Response: We fully agree with this reviewer since protection from fulminant hepatitis by perforin inhibition is the central claim of this manuscript. Therefore, we now provide survival curves regarding the genetic deficiency of prf1 (Figure 3J) and more importantly pharmacological inhibition of prf1 *in vivo* (Figure 7C). We can now demonstrate that prf1 inhibition leads to full protection and survival of infected mice.

3. Some further granularity on the effects in a few settings would be helpful. It is mentioned that the number of naïve T cells and the viral inoculum determined whether a severe or fulminant hepatitis occurred. It would be helpful to show these figures and ideally the difference in the pathology between the 2 settings.

Response: We now provide detailed data regarding the titration of the virus and the number of transferred CD8 T cells (Figure S1). Interestingly, only a high number of transferred OT-I cells leads to the death of mice, while lower numbers cause a similar pathology (loss of endothelial cells) yet to a lesser extent (Figure S1D/E). From a conceptual point of view, we think that there is a specific time window after infection during which cross-presenting liver endothelial cells are particularly vulnerable to T cell mediated attack. Importantly, low numbers of memory CD8 T cells are similarly able to cause severe pathology in our model (Figure 1A-D). This is likely based on their more rapid response kinetics compared to naïve CD8 T cells.

4. It would also be helpful to show a dose response curve for the pharmacological perforin inhibitor. Is there anything about the degree of perforin inhibition by this agent i.e. is the damage that is seen due to incomplete perforin inhibition or other mechanisms?

Response: The dosage we used for our mouse studies *in vivo* was deduced from independent pharmacokinetic studies (MTD, *in vivo* half life etc) performed in Dr Spicer's lab. To meet the reviewer's request, we have added a dose response curve of the inhibitor using purified murine perforin or intact NK cells *in vitro* (Figure S4). In comparison to the vehicle controls, and given the high efficiency of prf1 inhibition *in vitro* and the level of protection against T cell mediated liver failure *in vivo*, we conclude that mouse survival is related to maintaining an intact vascular endothelium, effectively limiting the extravasation of activated anti-Ova CTL into the liver parenchyma. Despite this, CTL infiltration into the parenchyma is not totally eliminated, nor do the transaminase levels return entirely to the normal range. Thus, perforin inhibition does not block 100% of hepatocyte cell death, but it does reduce it very substantially: to the point that the infected mice survive.

5. The authors expected that the number of virus-specific T cells pre-treatment would determine the outcome after infection (Fig 1I) but conclude this is not the case based on the correlation between T cell number and ALT. In truth, the curve looks like there is a fairly good correlation that is largely affected by 1 or possibly 2 outliers. Certainly not a major issue, but I would suggest that they not dismiss this association so strongly.

Response: We agree that we can't dismiss such an association and have modified our conclusion accordingly to: "We detected a mild correlation (Figure 1I), arguing that besides antigen-specific T cell numbers before infection also expansion and likely the presence of liver-resident T cells are critical factors regarding disease severity."

All changes in the manuscript text were highlighted in order to quickly assess the modifications.

Reviewers' comments:

Reviewer #1 (Remarks to the Author):

The authors have submitted a revised version of their manuscript. They have clearly tried to address the points by the reviewer and have indeed clarified certain points. Overall I feel the manuscript improved.

However, the major point is not yet addressed -- whether the perforin-mediated liver damage during fulminant viral hepatitis in this new mouse model can be applied as a main cause of acute fulminant viral hepatitis in human patients. The reviewer understand that it is not easy to obtain liver tissue from patients with fulminant hepatitis. However, since this issue is critical in asserting the clinical relevance of this mouse model, any experimental evidences for this should be included in this manuscript.

Point-by-point response to the reviewer's questions:

We thank the reviewers for taking the time to assess our manuscript in detail. We are happy to hear that most of the previous concerns of the reviewers have been sufficiently addressed.

We strongly believe that pharmacological perforin inhibition has the potential to treat patients with T cell and NK cell mediated pathologies and in particular fulminant hepatitis. From a pathophysiological perspective, we believe our study makes an important advance: it demonstrates that the breakdown of tolerance mechanisms that normally protect endothelial cells from CTL-mediated attack can be the major contributor to lymphocyte mediated organ failure, rather than T cell killing of virus-infected parenchymal cells, as has been assumed until now.

Below, we have addressed the concerns of reviewers in a point-by-point response:

Reviewer 1:

The authors have submitted a revised version of their manuscript. They have clearly tried to address the points by the reviewer and have indeed clarified certain points. Overall I feel the manuscript improved. However, the major point is not yet addressed - whether the perforin-mediated liver damage during fulminant viral hepatitis in this new mouse model can be applied as a main cause of acute fulminant viral hepatitis in human patients. The reviewer understand that it is not easy to obtain liver tissue from patients with fulminant hepatitis. However, since this issue is critical in asserting the clinical relevance of this mouse model, any experimental evidences for this should be included in this manuscript.

Response:

We can understand the reviewer's wish for us to mimic a genuine human pathology, since there is no animal model that will 'guarantee' a parallel response in virtually any human disease. Ultimately, all that any animal model can provide is a 'proof of concept' study that the approach is worthy of being tested in an appropriate clinical trial. The reviewers suggestion to obtain liver tissues from patients with fulminant hepatitis is, as pointed out by this reviewer, „not easy to obtain“. More importantly it will not allow us to address whether Prf1 mediated killing of liver endothelial cells is the critical mechanism that underlies fluminant hepatitis. It is well established that antiviral CD8 T cells in the context of hepatitis B and C express Prf1, yet causality can only be addressed in a proper clinical study. This request, however clearly goes beyond a reasonable revision of our manuscript.